# Sensory Receptor, Inflammatory, and Apoptotic Protein Expression in the Bladder Urothelium of Patients with Different Subtypes of Interstitial Cystitis/Bladder Pain Syndrome

**DOI:** 10.3390/ijms24010820

**Published:** 2023-01-03

**Authors:** Yuan-Hong Jiang, Jia-Fong Jhang, Lori A. Birder, Hann-Chorng Kuo

**Affiliations:** 1Department of Urology, Hualien Tzu Chi Hospital, Buddhist Tzu Chi Medical Foundation and Tzu Chi University, Hualien 970, Taiwan; 2Department of Medicine, Pharmacology and Chemical Biology, University of Pittsburgh, Pittsburgh, PA 15260, USA

**Keywords:** interstitial cystitis, bladder pain syndrome, biomarker, sensory protein

## Abstract

The aim of this study was to investigate the expression levels of sensory receptors, inflammatory proteins, and pro-apoptotic proteins in the urothelium of non-Hunner’s interstitial cystitis (NHIC) bladders of patients with different clinical and cystoscopic phenotypes. The urothelia from the bladders of 52 NHIC patients were harvested. The expression of sensory receptors, including TRPV1, TRPV4, TRPA1, H1-receptors, and sigma-1 receptors; the inflammatory proteins p38 and tryptase; and the pro-apoptotic proteins, such as caspase-3, BAD, and BAX in the urothelium, were investigated using immunohistochemistry and Western blotting. We compared the expression levels of these proteins in NHIC subtypes according to IC symptom scores, visual analog scores of bladder pain, maximal bladder capacity, glomerulation grades, and combined maximal bladder capacity and glomerulations after cystoscopic hydrodistention. The expression levels of TRPV1, TRPV4, sigma-1, P38, tryptase, caspase-3, and BAD were significantly increased in the urothelium of IC/BPS patients compared with the expression levels in the controls. TRPV1 was significantly associated with IC symptom severity. However, no significant differences in sensory receptor expression in the IC/BPS bladders with different bladder conditions were detected. Inflammatory and pro-apoptotic protein expression levels in the urothelium were similar among the IC/BPS subgroups. This study concluded that IC/BPS patients with frequency and bladder pain complaints have higher levels of urothelial sensory receptors, and inflammatory and pro-apoptotic proteins. The expression levels of these sensory receptors, inflammatory proteins, and pro-apoptotic proteins are not significantly different among IC/BPS bladders with different conditions.

## 1. Introduction

Interstitial cystitis/bladder pain syndrome (IC/BPS) is a chronic inflammatory disorder of the urinary bladder characterized by bladder pain and urinary frequency, urgency, nocturia, and sterile urine [1]. The clinical characteristics of IC/BPS are heterogeneous. The core symptom of IC/BPS is bladder pain, and different phenotypes are associated with different outcomes [2]. In addition to pain, patients with IC/BPS exhibit different clinical phenotypes, including urinary, psychosocial, organ-specific, infection, neurological/systemic, and tenderness systems [3]. Patients with IC/BPS have higher urine cytokine profiles which differ from the cytokine profiles of controls, suggesting the presence of chronic inflammation in the bladders of IC/BPS patients [4]. Urine cytokines levels significantly correlate with the clinical characteristics of the European Society of the Study of Interstitial Cystitis (ESSIC) type 2 IC/BPS patients, including symptom scores, visual analog scores (VASs) of pain severity, glomerulation grades, and maximal bladder capacity (MBC) [5].

Patients with IC/PBS exhibit predominant bladder pain or urgency frequency symptoms. Under cystoscopic hydrodistention, patients have reduced MBC under anesthesia, mild to severe glomerulations, and/or Hunner’s lesions, but some patients do not show these features [5]. Although urine or serum biomarkers, such as nerve growth factors, C-reactive proteins, antiproliferative factors, heparin-binding epidermal growth factors, and epidermal growth factors, are elevated in IC/BPS [6,7,8,9], not all patients with IC/BPS have elevated urinary biomarkers despite similar bladder pain symptoms and cystoscopic findings [4,8,9]. In addition, mast cell density does not appear to correlate with the duration of symptom amelioration after complete transurethral resections of Hunner’s lesions [10], and overall urine biomarkers do not associate with bladder biopsy findings [11]. Therefore, we speculated that the histological findings, urinary biomarkers, and clinical presentations of IC/BPS are heterogeneous. Among the different histopathological subtypes, increased bladder sensation plays a crucial role in the pathophysiology of IC/BPS [12].

The urothelium expresses many receptors typical of primary sensory neurons (e.g., the transient receptor potential (TRP) V1, TRPA1, and purinergic receptor P2X3), and neurons have high levels of sigma-1 receptors, which are involved in the modulation of urological pain. The sigma-1 receptor had been found to modulate somatic pain and be involved in the bladder pain in the cystitis [13]. Urothelial inflammation was attenuated in sigma-1-knockout mice compared with wild-type mice. In IC/BPS patients with bladders lacking remarkable inflammation and urothelial denudation, IC symptoms, such as pain and frequency urgency, are similar, but patients may have higher anxiety and stress status, resulting in bladder hypersensitivity and the associated symptoms [14,15]. Treatments targeting mucosal deficiency or chronic inflammation might be ineffective in these patients. Repeated psychological stress increases TRPV4 expression in the urothelium of rats [16]. Bladder capacity, voided volume, and inter-contraction intervals significantly increased following the administration of a TRPV4 antagonist, suggesting TRPV4 is involved in urinary bladder dysfunction disorders, such as IC/BPS. In addition, bladder distention triggers histamine release from mast cells, which induces peripheral and central hypersensitivity and may lead to increased IC/BPS, via interaction with the histamine H1 receptor and TRPV1 [17]. Due to the heterogeneity of this disease, treating IC/BPS by targeting the different pathophysiology is reasonable [18]. 

The purpose of this study was to investigate the expression of sensory receptors (TRPV1, TRPV4, TRPA1, H1-receptor, and sigma-1 receptors), inflammatory markers (p38 and tryptase), and pro-apoptotic markers (caspase-3, Bcl-2 antagonist of cell death (BAD), and BcL-2-associated X (BAX)) in the bladder urothelium of patients with non-Hunner’s IC (NHIC)/BPS with different cystoscopic phenotypes and histopathological subtypes. This study provides evidence that the pathophysiology of IC/BPS in patients with different cystoscopic IC features is related to altered sensory receptor expression in the bladder mucosa. The results of this study are clinically relevant in the decision-making for treatment strategies for patients with IC/BPS.

## 2. Results

The study included 52 female patients with IC/BPS and 6 controls. The mean age of patients with IC/BPS was significantly younger than the age of the controls. Table 1 shows the baseline symptom scores and VUDS parameters in patients with IC/BPS and the controls. The IC symptom indices and pain VAS, smaller bladder filling sensation, and cystometric bladder capacity were significantly higher in the IC/BPS patients compared with the controls, but the voiding efficiencies were similar between the groups.

The immunohistochemical staining of the location of the sensory receptors, TRPV1, TRPV4, TRPA1, P2X3, H-1, and sigma-1 receptors, is shown in Figure 1. These sensory receptors are located mainly in the bladder urothelium. The protein levels of the sensory receptors, inflammatory proteins, and pro-apoptotic proteins in the bladder tissue of the IC/BPS patients and controls, measured using Western blotting, are shown in Figure 2 and Appendix A. More IC/BPS patients showed higher expression levels of TRPV1, TRPV4, sigma-1 receptors, P38, tryptase, caspase 3, and BAD expressions than the controls.

Table 2 shows the sensory, inflammatory, and pro-apoptotic protein expressions in the urothelium of the IC/BPS patients and controls. The TRPV1, TRPV4, sigma-1 receptors, P38, tryptase, caspase 3, and BAD expression levels were higher in patients with IC/BPS compared with the controls. Except for TRPA1 and H-1, the other sensory receptors, and inflammatory and pro-apoptotic proteins show large or very large effect sizes in comparison with the controls. 

In the correlation analysis among all IC/BPs patients and controls, TRPV1 negatively associated with bladder volume (r = −0.294) and bladder compliance (r = −0.313), TRPV4 positively associated with voiding detrusor pressure (r = 0.278), and P2X3 negatively associated with the first sensation of bladder filling (r = −0.261). However, H-1 and sigma-1 did not correlate with any of the VUDS parameters. (Data not shown).

Only TRPV1 expression was significantly associated with the O’Leary–Sant symptom score (OSS), interstitial cystitis symptom index (ICSI), and interstitial cystitis problem index (ICPI). No significant associations of the other sensory receptors with IC symptom severity were detected. Significant positive correlations between TRPV1 and BAD, P2X3 and BAD, sigma-1 and caspase-3, and sigma-1 and BAD were detected. Negative correlations were detected between TRPV4 and tryptase, H-1 and P38, and H-1 and tryptase (Table 3). The scatter plots of the significant correlations between the expressions of sensory receptors and urodynamic parameters, inflammatory proteins, and pro-apoptotic proteins are shown in Appendix A. 

Table 4, Table 5, Table 6 and Table 7 show the expression levels of the sensory receptors, inflammatory proteins, and pro-apoptotic proteins in the bladders of IC/BPS patients and controls, according to different grades of glomerulation (Table 4), different MBC after hydrodistention (Table 5), different bladder pain VASs (Table 6), and different combinations of MBC and glomerulation grades (Table 7). The expression levels of the sensory receptors and inflammatory and pro-apoptotic proteins in the NHIC subtypes were significantly higher than the expression levels in the controls, but no significant differences in these receptors and protein levels were detected among NHIC patients with different bladder conditions.

## 3. Discussion

The expression levels of the sensory receptors, TRPV1, TRPV4, and sigma-1; the inflammatory proteins, P38 and tryptase; and the pro-apoptotic proteins, caspase-3 and BAD, were significantly increased in the urothelium of IC/BPS patients compared with the expression levels in the urothelium of the controls. The expression levels of several of these sensory receptors and inflammatory proteins significantly correlated with each other. However, the expression of the sensory receptors did not significantly differ in the IC/BPS bladders with different bladder conditions, such as MBC, glomerulation, and bladder pain severity. The sensory receptor expression levels were significantly associated with reduced first sensation of bladder filling and bladder capacity, and TRPV1 was significantly associated with IC symptom severity. Nevertheless, the inflammatory and pro-apoptotic protein expression levels in the urothelium were similar among IC/BPS patients with different bladder conditions.

The bladder urothelium expresses many receptors typical of primary sensory neurons (e.g., TRPV1, TRPA1, and P2X3), and neurons express high levels of sigma-1 receptors [13,19]. The expression of sigma-1 receptors in the urothelium may have functional relevance in bladder disorders, especially in cyclophosphamide-induced cystitis. The results of this study demonstrate that sigma-1 receptors are highly expressed in the urothelium of patients with IC/BPS, and the high expression of sigma-1 receptors is associated with increased expression levels of the pro-apoptotic proteins, caspase-3 and BAD. In contrast, the expression of the sensory receptor H-1 did not increase significantly in the bladder urothelium of patients with IC/BPS. A previous study demonstrated that H-1 receptors play a role in the mechanic distention-related hypersensitivity of bladder-innervating sensory neurons such as TRPV1 [17]. In this study, the expression of H-1 receptors was negatively associated with inflammatory protein expression (P38 and tryptase), suggesting that increases in these inflammatory proteins in the urothelium of patients with IC/BPS are not related to H-1 receptors. Increased bladder sensation, pain, and altered bladder function during bladder distention in patients with IC/BPS is likely initiated by inflammation, increased urothelial apoptosis, and increased urothelial permeability [20].

Multiple TRP channels (TRPV1, TRPV4, and TRPA1) are expressed in the bladder with specific tissue distributions in the lower urinary tract. These TRP channels are implicated in bladder disorders, including IC/BPS. The TRPV4 channels are strong candidates for mechanosensors in the urinary bladder, and TRPV4 antagonists are promising therapeutic agents for OAB [21]. TRPV1 and TRPA1 are required for inflammatory visceral hyperalgesia [22]. In one study, the percentage of bladder afferents expressing functional TRPA1, but not TRPV1, was significantly elevated for 7 days after cyclophosphamide treatment. However, in the present study, TRPV1 and TRPV4, but not TRPA1, were significantly increased in the bladder urothelium of patients with IC/BPS. Furthermore, the expression levels of these TRP channels were not significantly different among IC/BPS patients with different bladder conditions. Moreover, TRPV1 expression was significantly associated with IC symptom severity, suggesting that these TRP channels are downstream responses to inflammation but are not involved in the primary pathophysiology of increased bladder sensation in IC/BPS patients. 

IC/BPS symptoms include urinary urgency, frequency, and pain, indicating a key role for hypersensitivity in bladder-innervating sensory neurons. This hypersensitivity translates to increased sensory input and activation in the spinal cord, which may underlie the symptoms of bladder hypersensitivity and pain experienced in IC/BPS patients [17]. These sensory receptors may be activated at a lower threshold of bladder mucosal stretching and urinary potassium concentration in response to the higher inflammatory conditions in IC/BPS patients. Among the apoptotic and inflammatory proteins, BAD, BAX, caspase 3, p53, p27, and tumor necrosis factor-α are expressed at higher levels in the bladders of IC/BPS patients compared with the controls [19]. The bladder mucosa exhibits urothelial thinning and increased immunoreactivity to caspase-3 and BAX, suggesting that chronic sympathetic stimulation may contribute to IC/BPS pathophysiology [16]. In this study, several inflammatory proteins (P38 and tryptase) and several pro-apoptotic proteins (caspase-3 and BAD) were significantly increased in the bladders of patients with IC/BPS. Changes in the expression of both inflammatory and apoptosis proteins suggest a network exists in IC/PBS bladders. 

Our recent study demonstrated that patients with IC/BPS can be stratified according to the cystoscopic hydrodistention findings. The clinical characteristics, symptoms, urodynamic parameters, and findings of cystoscopic hydrodistention are distinct between patients with HIC and NHIC [5]. Among the four NHIC subgroups, patients with glomerulation grade 0 or 1 and MBC ≥ 760 mL exhibited better urodynamic parameters and the highest rate of satisfactory treatment outcome [5]. However, in this study, the expression of the sensory receptor, sigma-1, was elevated but similar among the NHIC subgroups. Bladder mucosal inflammatory and apoptotic protein levels did not significantly differ among different IC/BPS bladder conditions, such as different grades of glomerulations, different MBC, different bladder pain VASs, and different combinations of MBC and glomerulations. Thus, inflammation and apoptotic processes may be similar among IC/BPS bladders; however, the clinical IC/BPS phenotypes may differ according to the individual innate immunity of patients. Nevertheless, urine inflammatory and oxidative stress biomarker levels were significantly elevated in NHIC patients with glomerulation grade ≥ 2 and in NHIC patients with MBC ≤ 700 [23]. These findings suggest that NHIC bladders exhibit inflammation and mucosal apoptosis characteristics, which cause bladder hyperalgesia and elevated urinary biomarkers. 

Positive bladder histopathological findings are associated with a smaller bladder capacity, denuded urothelium, increased inflammation, reduced urothelial proliferative function, and increased bladder sensation [12,24,25]. These results suggest that positive bladder histopathological findings are associated with clinical symptoms and may have an impact on treatment outcomes among patients with IC/BPS [26]. The current study also shows that inflammatory cell infiltration increased significantly more than urothelium denudation, and inflammatory cell infiltration and urothelium denudation significantly correlated [27]. These results confirm that sensory receptors are activated in association with increased inflammatory proteins and pro-apoptotic proteins in the bladders of IC/BPS patients.

Psychological (as well as physical) stressors exacerbate the symptoms of IC/BPS. Stress and urinary urgency are significantly related [14]. In our previous study, we demonstrated the expression of the stress-response receptor, corticotropin-releasing hormone receptor (CRHR) in the bladders of IC/BPS patients is altered and may contribute to sensory dysfunction in IC/BPS patients [15], suggesting that IC/BPS bladder disorder is closely associated with stress. Increased bladder sensory receptors in IC/BPS patients with higher anxiety status might be due to the activation of the limbic system of the brain, resulting in mild bladder inflammation but higher clinical symptoms [20]. Our previous study demonstrated higher urinary inflammatory protein levels and oxidative biomarkers in the IC/BPS with lower MBC and a higher glomerulation grade [23]. However, we did not detect differences in the sensory receptor expression levels in IC/BPS patients with different MBC, glomerulations, and bladder pain severity. Thus, bladder inflammatory proteins, sensory receptors, and urinary biomarkers in IC/BPS bladders may be diverse.

The limitation of this study is the different mean age between the IC/BPS patients and control women. It is not easy to obtain normal bladder tissues from women with completely normal urinary tract function which is confirmed using video urodynamic study. We cannot be certain that the differences in the sensory receptors of IC/BPS patients are not caused by age. However, a previous study revealed that the ageing process increases the sensory receptors and neuropeptide expressions from the bladder urothelium, which results in increased afferent mechanosensitivity and increased smooth muscle contractility. Therefore, the higher age in the control women might not affect the results of this study [28,29]. In addition, this is a retrospective analysis as we collected our previously biopsied bladder tissue for analysis. The purpose of this study was to investigate the sensory receptor expressions in the urothelium of patients with IC/BPS of different clinical phenotypes. Therefore, although the expression of sensory receptors, and inflammatory and pro-apoptotic proteins are significantly higher than those in the controls, the number of patients with different IC/BPS subgroups is small, which might cause the small effect size in the analysis of the expression of these proteins among different IC/BPS subgroups. Nevertheless, the results of this study are clinically relevant in the decision-making of treatment strategies for patients with IC/BPS.

## 4. Materials and Methods

Bladder biopsies from IC/BPS female patients involved in our previous studies were analyzed for this study [12,15]. Bladder biopsies were obtained after informed consent. Patients were clinically diagnosed with IC/BPS according to a video urodynamic study (VUDS), positive potassium chloride test confirmation, cystoscopic hydrodistention features [5], and histopathological analysis, as previously reported [12]. The patients were classified into different NHIC subgroups according to the MBC and glomerulations after cystoscopic hydrodistention, and Hunner’s IC (HIC). Bladder symptom severity was evaluated using the VAS for bladder pain, interstitial cystitis symptom index, interstitial cystitis problem index, and O’Leary–Sant symptom score. This study was approved by the ethics committee of our institute (IRB: TCGH 110-169-C, dated August/12/2021). Informed consent was waived due to the retrospective analysis.

All patients had not been treated with intravesical therapies such as botulinum toxin A or platelet-rich plasma. Patients with IC/BPS underwent endoscopic cold-cup biopsies of the bladder wall, including the mucosa and submucosa, during intravesical treatment, such as botulinum toxin injections or platelet-rich plasma injections, after cystoscopic hydrodistention. The diameter of each specimen was approximately 2 mm. Bladder biopsies from NHIC subjects were obtained from the posterior wall near the glomerulation hemorrhage. Bladder biopsies from HIC patients were obtained from sites with relatively intact mucosa near Hunner’s lesions. Endoscopic electrocauterization of the biopsy sites was performed to prevent bleeding. Histopathological analysis of the IC/BPS bladder urothelium was performed as previously reported [12]. Bladder biopsy specimens from patients with IC/BPS were stained with hematoxylin and eosin. A single pathologist blinded to the clinical results (Hsu YH) reviewed all bladder histopathological findings and excluded the presence of carcinoma in situ. Six women (aged 55 to 78 years) with genuine stress urinary incontinence and VUDS-verified normal lower urinary tract function served as controls and donated their bladder mucosa specimens during suburethral sling surgery. These control women had participated in our previous clinical study, and informed consent had been obtained.

### 4.1. Immunohistochemistry Study

Urinary bladder biopsy specimens from patients with IC/BPS were embedded in OCT medium and stored at −80 °C. Four 5-μm sections per specimen were obtained using a cryostat and collected on new silane III-coated slides (Muto Pure Chemicals Co. Ltd., Tokyo, Japan). The sections were post-fixed in acetone at −20 °C, blocked with rabbit serum, and incubated overnight at 4 °C. Immunohistochemical staining of biopsies from NHIC bladders was performed using primary antibodies to TRPV1, TRPV4, TRPA1, P2X3, H1-receptor, sigma-1 receptors, tryptase, p38, caspase-3, BAD, and BAX. GAPDH was used as a normalizing protein for quantification. The immunohistochemistry study was performed using the UltraVision Quanto Detection System HRP DAB (ThermoScientific, Cheshire, UK, catalog number: TL-060-QHD). Slides were first treated with hydrogen peroxide block reagent (ThermoScientific) and rinsed with PBS. Following that, Ultra V Block reagent (ThermoScientific) was used to block nonspecific binding. The slides were subsequently incubated with primary antibodies including TRPV1 (dilution 1:500, from GeneTex, CA, USA, catalog number: GTX10296), TRPV4 (dilution 1:200, from GeneTex, CA, USA, catalog number: GTX10296), TRPA1 (dilution 1:500, from GeneTex, CA, USA, catalog number: GTX54765), P2X3 (dilution 1:200, from GeneTex, CA, USA, catalog number: GTX10269), H-1 (dilution 1:25, from Santa Cruz Biotechnology, CA, USA, catalog number: sc-374621), and sigma-1 (dilution 1:50, from Santa Cruz Biotechnology, CA, USA, catalog number: sc-137075). The Primary Antibody Amplifier Quanto (ThermoScientific), HRP Polymer Quanto (ThermoScientific), DAB Quanto Chromogen, and DAB Quanto substrate (ThermoScientific) were used to visualize. The procedures were in accordance with our previous study. The methodology is described in previous studies [30,31,32]. We compared the urothelial sensory receptors, inflammatory proteins, and pro-apoptotic markers between IC/BPS and control bladders, and among different NHIC subtypes according to IC symptom scores, VASs of bladder pain, MBC, glomerulation grades, and combined MBC and glomerulations after cystoscopic hydrodistention.

### 4.2. Western Blotting

The bladder biopsy specimens from the IC/BPS and control patients were homogenized in liquid nitrogen and then lysed for 10 min on ice using Hytra tissue protein extraction reagent (Hycell Biotechnology Inc., Taipei City, Taiwan). The extraction solution was supplemented with a protease inhibitor cocktail (Roche Diagnostics, Mannheim, Germany) and a phosphatase inhibitor cocktail (Roche Diagnostics). Proteins were separated using electrophoresis on 10% Tris-glycine gel. After gel electrophoresis, the protein blots were transferred to 0.2-μm PVDF membranes using 3% skim milk as blocking buffer for 1 hour, then adding TRPA1 (dilution 1:8000, from Sigma-Aldrich, St. Louis, MO, USA, catalog number: SAB1411593), TRPV1 (dilution 1:30,000, from Sigma-Aldrich, St. Louis, MO, USA, catalog number: SAB3501027), TRPV4 (dilution 1:10,000, from Sigma-Aldrich, St. Louis, MO, USA, catalog number: SAB2104243), P2X3 (dilution 1:1000, from GeneTex, CA, USA, catalog number: GTX100118), P38 (dilution 1:500, from Cell Signaling Technology, catalog number: #9212S),Tryptase (dilution 1:500, from Millipore Corporation, Burlington, MA, USA, catalog number: MAB1222), Caspase3 (dilution 1:1000, from Cell Signaling Technology, catalog number: #9662), BAX (dilution 1:500, from Cell Signaling Technology, catalog number: #2772S), BAD (dilution 1:500, from Cell Signaling Technology, catalog number: #9292S), H-1 (dilution 1:500, from Santa Cruz Biotechnology, CA, USA, catalog number: sc-374621), sigma-1 (dilution 1:500, from Santa Cruz Biotechnology, CA, USA, catalog number: sc-137075) as primary antibodies and GAPDH (Cell Signaling Technology, catalog number: #2118S) as positive control. The membranes were incubated overnight at 4 °C; after incubation, they were washed with TBST 4 times each for 10 min. The secondary antibody (goat anti-rabbit IgG-HRP; 1:5000, Millipore Corporation, USA, catalog number: AP132P) was then applied. The membranes were finally probed with enhanced chemiluminescence reagent (ECL; Millipore Corporation, USA) and exposed to X-ray films. The scanned film after gel electrophoresis was quantified using a gel documentation system (Quantity One Version 4.6.2, Bio-Rad Laboratories, Hemel Hempstead, Hertfordshire, UK). GAPDH was used as normalizing protein for the quantification. All of the bladder samples from all patients were analyzed using identical techniques. [15] The intensity of each protein was determined then normalized to GAPDH as the loading control. Data are presented as mean ± standard deviation of the relative density of sensory, inflammatory, and pro-apoptotic protein expression to GAPDH, between patients with IC/BPS and controls, and among different IC/BPS subgroups and controls.

### 4.3. Statistical Analysis

Descriptive statistics was expressed as means ± standard deviation (SD) or percentages. Differences in expressions of sensory receptors between IC/BPS and controls were analyzed using Student’s *t*-test, and among multiple IC/BPS subgroups and the controls were analyzed using the Kruskal–Wallis test (non-parametric ANOVA). Effect size was calculated using SPSS, and Cohen’s d and Eta squared values were presented for 2-group and multiple group comparisons, respectively, in the statistical analysis. The magnitudes of effect size were 0.20, 0.50, 0.8, and 1.2 for small, moderate, large, and very large effects in the *t*-test, respectively [33]. In multiple group comparison, the effect size was 0.10, 0.25, and 0.40 representing small, medium, and large effects, respectively [34]. The post hoc tests based on the Mann–Whitney U test with Bonferroni’s correction for controlling overall type-1 errors were performed by comparing the differences between two subgroups. Pearson correlation coefficients were calculated between the sensory receptors and inflammatory protein expressions as well as clinical and urodynamic variables. The Statistical Package for the Social Sciences software for Windows version 20.0 (IBM Corp., Armonk, NY, USA) was used for statistical analysis, and *p* values of <0.05 were considered statistically significant.

## 5. Conclusions

Sensory receptors (TRPV1, TRPV4, and sigma-1 receptors), inflammatory proteins (tryptase and P38), and pro-apoptotic proteins (caspase-3 and BAD) are significantly elevated in the bladders of IC/BPS patients. Elevated sensory receptor levels are associated with lower bladder sensation (P2X3), smaller volume (TRPV1), and lower bladder compliance (TRPV4). Although TRPV1 expression is significantly associated with IC symptom severity, these sensory receptors, inflammatory proteins, and pro-apoptotic protein levels are not significantly different among IC/BPS bladders with different conditions. IC/BPS patients with frequency and bladder pain complaints have higher urothelial sensory receptor, inflammatory, and pro-apoptotic protein expression levels, which might be downstream of chronic inflammation.

## Figures and Tables

**Figure 1 ijms-24-00820-f001:**
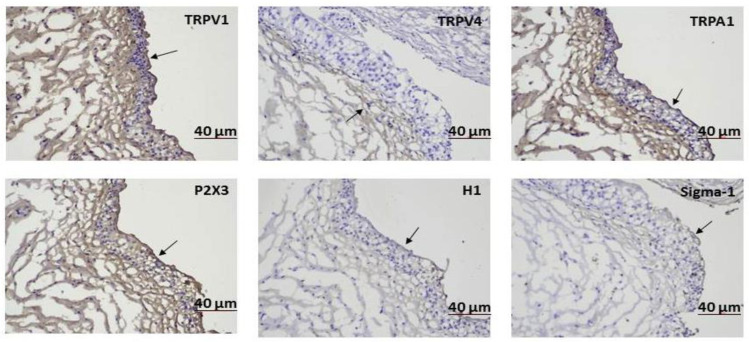
Immunohistochemistry staining of sensory receptors in the bladder urothelium of patients with interstitial cystitis/bladder pain syndrome. Arrows indicate the expression of these sensory receptors in the bladder urothelium.

**Figure 2 ijms-24-00820-f002:**
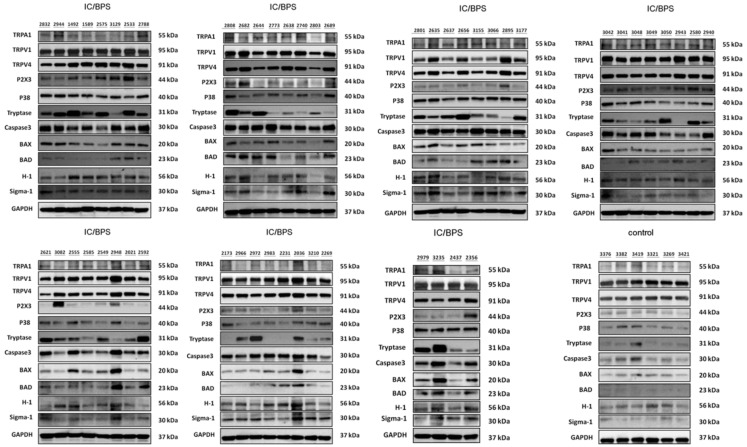
Western blots of bladder urothelial sensory receptors (TRPA1, TRPV1, TRPV4, P2X3, H-1, and sigma-1), and inflammatory (p38, tryptase) and pro-apoptotic proteins (caspase-3, BAX, BAD) in the bladder urothelium of IC/BPS patients and controls. More IC/BPS patients showed higher expression levels of TRPV1, TRPV4, sigma-1 receptors, P38, tryptase, caspase 3, and BAD expressions than controls. Molecular weight (kD) is indicated to the right of the blot. The GAPDH was blotted and used as a loading control.

**Table 1 ijms-24-00820-t001:** The interstitial cystitis symptom score and video urodynamic parameters in interstitial cystitis/bladder pain syndrome patients and controls.

VUDS Parameters	IC/BPS (*n* = 52)	Control (*n* = 6)	*p* Value	Effect Size (Cohen’s d) *
Age (years)	54.6 ± 12.0	67.8 ± 4.5	<0.001	
OSS	20.77 ± 7.82	0		
CSI	10.65 ± 4.4	0		
ICPI	10.12 ± 3.8	0		
VAS	4.31 ± 2.8	0		
Pdet (cm H_2_O)	21.22 ± 14.06	16.4 ± 8.91	0.458	0.495
Qmax (mL/s)	9.90 ± 5.85	17.6 ± 4.93	0.006	1.423
Volume (mL)	192.61 ± 119.91	434.6 ± 191.57	<0.001	1.514
PVR (mL)	67.14 ± 118.83	90 ± 174.64	0.696	0.153
FSF (mL)	121.44 ± 54.02	181.6 ± 52.88	0.021	1.125
FS (mL)	189.27 ± 76.56	361.6 ± 106.37	<0.001	1.860
Compliance	65.54 ± 47.19	108.24 ± 50.74	0.060	0.871
BCI	70.51 ± 31.99	104.4 ± 19.65	0.025	1.276
CBC (mL)	255.57 ± 120.51	524.6 ± 42.89	<0.001	2.974
cQmax	0.62 ± 0.33	0.77 ± 0.20	0.331	0.549
VE	0.75 ± 0.33	0.82 ± 0.34	0.638	0.209

VUDS: video urodynamic study, IC/BPS: interstitial cystitis/bladder pain syndrome, OSS: O’Leary– Sant symptom score, ICSI: interstitial cystitis symptom index, ICPI: interstitial cystitis problem index, VAS: visual analog score of pain, Pdet: detrusor pressure, Qmax: maximum flow rate, PVR: post-void residual, FSF: first sensation of filling, FS: full sensation, BCI: bladder contractility index, CBC: cystometric bladder capacity, cQmax: corrected maximum flow rate, VE: voiding efficiency. * The magnitudes of effect size are 0.20, 0.50, 0.8, and 1.2 for small, moderate, large, and very large effects, respectively.

**Table 2 ijms-24-00820-t002:** The sensory, inflammatory, and pro-apoptotic protein expression levels in the urothelium of patients with interstitial cystitis/bladder pain syndrome and controls.

Bladder Protein	IC/BPS (*n* = 52)	Control (*n* = 6)	*p* Value	Effect Size * (Cohen’s d)
TRPV1	1.01 ± 0.30	0.79 ± 0.09	0.001	0.99
TRPV4	0.84 ± 0.19	0.67 ± 0.05	0.036	1.22
TRPA1	0.12 ± 0.06	0.14 ± 0.05	0.282	0.36
P2X3	0.24 ± 0.15	0.14 ± 0.04	0.128	0.91
H-1	0.34 ± 0.15	0.25 ± 0.08	0.145	0.75
Sigma-1	0.38 ± 0.19	0.15 ± 0.07	<0.001	1.61
P38	0.39 ± 0.20	0.14 ± 0.07	<0.001	1.67
Tryptase	0.68 ± 0.48	0.11 ± 0.08	<0.001	1.66
Caspase3	0.95 ± 0.35	0.25 ± 0.11	<0.001	2.70
BAX	0.33 ± 0.18	0.22 ± 0.07	0.134	0.81
BAD	0.28 ± 0.21	0.02 ± 0.01	<0.001	1.75

Abbreviations: as in text. Data are presented as mean ± standard deviation of the relative density of sensory, inflammatory, and pro-apoptotic protein expression to GAPDH, using Western blotting analysis, in the bladder tissue of IC/BPS and controls. * The magnitudes of effect size are 0.20, 0.50, 0.8, and 1.2 for small, moderate, large, and very large effects, respectively.

**Table 3 ijms-24-00820-t003:** The correlation between sensory receptor density and the clinical symptoms, inflammatory and pro-apoptotic protein expressions in the urothelium of patients with interstitial cystitis/bladder pain syndrome.

IC/BPS	TRPV1	TRPV4	TRPA1	P2X3	H-1	Sigma-1
OSS	0.398 **	0.168	0.007	0.153	0.105	0.029
ICSI	0.405 **	0.121	0.031	0.112	0.031	0.008
ICPI	0.349 *	0.205	−0.022	0.184	0.180	0.050
VAS	0.003	−0.037	−0.001	−0.049	−0.041	−0.072
P38	−0.299 *	0.002	0.228	−0.088	−0.315 *	−0.049
Tryptase	−0.252	−0.470 **	0.204	−0.064	−0.321 *	−0.104
Caspase 3	0.261	−0.183	0.110	−0.029	0.038	0.521 **
BAX	−0.026	0.039	0.262	−0.073	0.042	0.121
BAD	0.519 **	0.018	0.097	0.276 *	0.239	0.474 **

Abbreviations: as in text, * *p* < 0.05, ** *p* < 0.005. Data are presented as mean ± standard deviation of the relative density of sensory, inflammatory, and pro-apoptotic protein expression to GAPDH, using Western blotting analysis, in the bladder tissue of IC/BPS and controls.

**Table 4 ijms-24-00820-t004:** Sensory, inflammatory, and pro-apoptotic protein expression levels in the urothelium of patients with interstitial cystitis/bladder pain syndrome with different grades of glomerulations and controls.

Bladder Protein	1. Gr 0–1 (*n* = 13)	2. Gr 1 (*n* = 18)	3. Gr 2 (*n* = 16)	4. Gr 3,4 (*n* = 5)	5. Control (*n* = 6)	*p* Value	Effect Size Eta Squared *	Post Hoc
Age (years)	55.5 ± 13.3	57.3 ± 11.5	48.1 ± 9.73	62.8 ± 9.31	67.8 ± 4.45	0.003	0.254	123 vs. 5, 3 vs. 24
TRPV1	1.01 ± 0.22	1.02 ± 0.29	0.95 ± 0.29	1.20 ± 0.52	0.79 ± 0.09	0.358	0.102	
TRPV4	0.83 ± 0.20	0.84 ± 0.20	0.85 ± 0.19	0.83 ± 0.15	0.67 ± 0.05	0.359	0.078	
TRPA1	0.11 ± 0.06	0.11 ± 0.05	0.12 ± 0.05	0.16 ± 0.07	0.14 ± 0.05	0.294	0.087	
P2X3	0.20 ± 0.11	0.25 ± 0.19	0.23 ± 0.10	0.31 ± 0.26	0.14 ± 0.04	0.476	0.077	
H-1	0.39 ± 0.18	0.34 ± 0.13	0.30 ± 0.12	0.36 ± 0.24	0.25 ± 0.08	0.316	0.084	
Sigma-1	0.41 ± 0.21	0.36 ± 0.18	0.37 ± 0.18	0.48 ± 0.22	0.15 ± 0.07	0.037	0.172	1234 vs. 5
P38	0.37 ± 0.17	0.40 ± 0.23	0.40 ± 0.18	0.40 ± 0.24	0.14 ± 0.07	0.074	0.147	
Tryptase	0.74 ± 0.48	0.80 ± 0.59	0.50 ± 0.35	0.69 ± 0.37	0.11 ± 0.08	0.008	0.191	123 vs. 5
Caspase3	0.96 ± 0.33	1.01 ± 0.35	0.78 ± 0.32	1.25 ± 0.21	0.25 ± 0.11	<0.001	0.414	12 vs. 5, 3 vs. 45
BAX	0.37 ± 0.20	0.34 ± 0.16	0.26 ± 0.13	0.43 ± 0.25	0.22 ± 0.07	0.117	0.128	
BAD	0.27 ± 0.22	0.31 ± 0.23	0.19 ± 0.13	0.44 ± 0.30	0.02 ± 0.01	0.024	0.238	123 vs. 5

Abbreviations: as in text, Gr: grade of glomerulation after cystoscopic hydrodistention. Data are presented as mean ± standard deviation of the relative density of sensory, inflammatory, and pro-apoptotic protein expression to GAPDH, using Western blotting analysis, in the bladder tissue of IC/BPS and controls. * The magnitudes of effect size are 0.10, 0.25, and 0.40 and represent small, medium, and large effect sizes, respectively.

**Table 5 ijms-24-00820-t005:** Sensory, inflammatory, and pro-apoptotic protein expression levels in the urothelium of patients with interstitial cystitis/bladder pain syndrome with maximal bladder capacity and controls.

Bladder Tissue	1. MBC < 700 mL (*n* = 24)	2. MBC = 700–900 mL (*n* = 17)	3. MBC ≥ 900 mL (*n* = 11)	4. Control (*n* = 6)	*p* Value	Effect Size Eta Squared *	Post hoc
Age (years)	57.2 ± 11.5	52.2 ± 11.7	52.6 ± 13.4	67.8 ± 4.5	0.031	0.150	123 vs. 4
TRPV1	0.99 ± 0.31	1.03 ± 0.32	1.02 ± 0.28	0.79 ± 0.09	0.371	0.056	
TRPV4	0.83 ± 0.17	0.87 ± 0.17	0.81 ± 0.25	0.67 ± 0.05	0.160	0.087	
TRPA1	0.11 ± 0.04	0.12 ± 0.07	0.13 ± 0.07	0.14 ± 0.05	0.516	0.044	
P2X3	0.23 ± 0.16	0.25 ± 0.17	0.24 ± 0.13	0.14 ± 0.04	0.480	0.044	
H-1	0.33 ± 0.15	0.36 ± 0.16	0.33 ± 0.16	0.25 ± 0.08	0.478	0.045	
Sigma-1	0.41 ± 0.20	0.39 ± 0.19	0.31 ± 0.14	0.15 ± 0.07	0.003	0.173	123 vs. 4
P38	0.38 ± 0.20	0.37 ± 0.19	0.45 ± 0.20	0.14 ± 0.07	0.020	0.166	123 vs. 4
Tryptase	0.68 ± 0.49	0.62 ± 0.43	0.79 ± 0.55	0.11 ± 0.08	0.020	0.145	123 vs. 4
Caspase3	0.98 ± 0.34	0.88 ± 0.35	0.99 ± 0.38	0.25 ± 0.11	<0.001	0.313	123 vs. 4
BAX	0.32 ± 0.19	0.33 ± 0.17	0.36 ± 0.17	0.22 ± 0.07	0.475	0.045	
BAD	0.30 ± 0.22	0.24 ± 0.19	0.28 ± 0.24	0.02 ± 0.01	0.016	0.150	123 vs. 4

Abbreviations: as in text, MBC: maximal bladder capacity after cystoscopic hydrodistention. Data are presented as mean ± standard deviation of the relative density of sensory, inflammatory, and pro-apoptotic protein expression to GAPDH, using Western blotting analysis, in the bladder tissue of IC/BPS and controls. * The magnitudes of effect size are 0.10, 0.25, and 0.40 and represent small, medium, and large effect sizes, respectively.

**Table 6 ijms-24-00820-t006:** Sensory, inflammatory, and pro-apoptotic protein expression levels in the urothelium of patients with interstitial cystitis/bladder pain syndrome with different bladder pain severity and controls.

Bladder Protein	1. VAS ≥ 6 (*n* = 22)	2. VAS <6 (*n* = 30)	3. Control (*n* = 6)	*p* Value	Effect Size Eta Squared *	Post hoc
Age (years)	55.1 ± 10.8	54.2 ± 12.9	67.8 ± 4.45	0.035	0.115	12 vs. 3
TRPV1	0.99 ± 0.27	1.03 ± 0.33	0.79 ± 0.09	0.079	0.056	
TRPV4	0.85 ± 0.20	0.83 ± 0.18	0.67 ± 0.05	0.107	0.078	
TRPA1	0.11 ± 0.05	0.12 ± 0.06	0.14 ± 0.05	0.268	0.047	
P2X3	0.23 ± 0.13	0.24 ± 0.17	0.14 ± 0.04	0.316	0.041	
H-1	0.34 ± 0.15	0.34 ± 0.15	0.25 ± 0.08	0.349	0.038	
Sigma-1	0.37 ± 0.17	0.40 ± 0.20	0.15 ± 0.07	0.002	0.142	12 vs. 3
P38	0.40 ± 0.19	0.39 ± 0.20	0.14 ± 0.07	0.002	0.144	12 vs. 3
Tryptase	0.70 ± 0.52	0.67 ± 0.46	0.11 ± 0.08	0.003	0.131	12 vs. 3
Caspase3	0.96 ± 0.34	0.94 ± 0.36	0.25 ± 0.11	<0.001	0.300	12 vs. 3
BAX	0.39 ± 0.19	0.29 ± 0.15	0.22 ± 0.07	0.027	0.124	1 vs. 2, 1 vs. 3
BAD	0.30 ± 0.22	0.26 ± 0.22	0.02 ± 0.01	0.002	0.140	12 vs. 3

Abbreviations: as in text, VAS: visual analog score of bladder pain (from 0–10). Data are presented as mean ± standard deviation of the relative density of sensory, inflammatory, and pro-apoptotic protein expression to GAPDH, using Western blotting analysis, in the bladder tissue of IC/BPS and controls. * The magnitudes of effect size are 0.10, 0.25, and 0.40 and represent small, medium, and large effect sizes, respectively.

**Table 7 ijms-24-00820-t007:** Sensory, inflammatory, and pro-apoptotic protein expression levels in the urothelium of patients with interstitial cystitis/bladder pain syndrome with different bladder conditions during hydrodistention and controls.

Bladder Protein	1. All MBC Gr ≥ 2 (*n* = 21)	2. MB >700 mL Gr 0,1(*n* = 13)	3. MBC ≦ 700 mL Gr 0,1 (*n* = 18)	4. Control (*n* = 6)	*p* Value	Effect Size Eta Squared *	Post hoc
Age	51.6 ± 11.4	50.6 ± 13.3	60.9 ± 9.25	67.8 ± 4.45	0.001	0.247	1 vs. 34, 2 vs. 34
TRPV1	1.01 ± 0.36	1.10 ± 0.29	0.95 ± 0.22	0.79 ± 0.09	0.092	0.088	
TRPV4	0.84 ± 0.18	0.81 ± 0.22	0.86 ± 0.18	0.67 ± 0.05	0.171	0.088	
TRPA1	0.13 ± 0.06	0.12 ± 0.06	0.10 ± 0.04	0.14 ± 0.05	0.241	0.074	
P2X3	0.25 ± 0.15	0.25 ± 0.15	0.22 ± 0.17	0.14 ± 0.04	0.429	0.050	
H-1	0.32 ± 0.15	0.34 ± 0.15	0.37 ± 0.15	0.25 ± 0.08	0.332	0.061	
Sigma-1	0.40 ± 0.19	0.34 ± 0.16	0.41 ± 0.21	0.15 ± 0.07	0.009	0.156	123 vs. 4
P38	0.40 ± 0.19	0.45 ± 0.18	0.34 ± 0.21	0.14 ± 0.07	0.013	0.180	123 vs. 4
Tryptase	0.55 ± 0.36	0.76 ± 0.58	0.79 ± 0.52	0.11 ± 0.08	0.008	0.179	123 vs. 4
Caspase3	0.89 ± 0.36	1.07 ± 0.36	0.93 ± 0.32	0.25 ± 0.11	0.000	0.331	123 vs. 4
BAX	0.30 ± 0.18	0.39 ± 0.15	0.33 ± 0.19	0.22 ± 0.07	0.221	0.078	
BAD	0.25 ± 0.21	0.31 ± 0.26	0.29 ± 0.20	0.02 ± 0.01	0.020	0.145	123 vs. 4

Abbreviations: as in text, MBC: maximal bladder capacity, Gr: grade of glomerulation after cystoscopic hydrodistention. Data are presented as mean ± standard deviation of the relative density of sensory, inflammatory, and pro-apoptotic protein expression to GAPDH, using Western blotting analysis, in the bladder tissue of IC/BPS and controls. * The magnitudes of effect size are 0.10, 0.25, and 0.40 and represent small, medium, and large effect sizes, respectively.

## Data Availability

Data in this study can be obtained with the request and permission to the corresponding author.

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
