# Peer review of "Sensory Receptor, Inflammatory, and Apoptotic Protein Expression in the Bladder Urothelium of Patients with Different Subtypes of Interstitial Cystitis/Bladder Pain Syndrome"

_ijms, 2023, doi:10.3390/ijms24010820_

Round 1
Reviewer 1 Report
In the manuscript entitled „Sensory receptor, inflammatory, and apoptotic protein expression in the bladder urothelium of patients with different subtypes of interstitial cystitis/bladder pain syndrome,” authors where analyzing different inflammation and apoptotic parameters as well as sensory receptors in IC/BPS patients. The idea behind this study was good. Unfortunately, the manuscript has been poorly written. The results are unclear and the quality of the data is poor, which can lead to incorrect conclusions. The Materials and Methods section is inadequately written, and the main method of the manuscript is missing. The quality of IHC is shown in Figure 1. is quite bad. It has a large amount of background staining with unambiguously positive cells. The main results are shown in Fig. 2. Representative blots were also of poor quality, and it was difficult to compare samples from the left and right blot panels. GAPDH, a housekeeping protein, is overexposed and cannot be used to normalize the expression of other proteins. It is unclear how the blots were analyzed and how they obtained the data in Table 2. The full blots are not provided in the Supplementary Material. Samples from the control and IC/BPS patients should be run on the same gels for comparison. It should be at least two control samples for each gel with IC/BPS patient samples. The statistical analysis in table 3-7 is correct, although the data used in the tables are questionable, as explained before. The discussion is long, with many irrelevant details, and the main points of the manuscript have not been discussed. Unfortunately, I cannot see how this work meets the standards of a high-impact journal, such as the International Journal of Molecular Sciences.Author Response
Reviewer #1:
In the manuscript entitled „Sensory receptor, inflammatory, and apoptotic protein expression in the bladder urothelium of patients with different subtypes of interstitial cystitis/bladder pain syndrome,” authors were analyzing different inflammation and apoptotic parameters as well as sensory receptors in IC/BPS patients. The idea behind this study was good. Unfortunately, the manuscript has been poorly written. The results are unclear and the quality of the data is poor, which can lead to incorrect conclusions.
Reply: Thank you for the comment. The purpose of this study was to investigate the sensory receptor expressions in the urothelium of patients with IC/BPS of different clinical phenotypes. We hypothesized that patients with IC/BPS and having nearly normal bladder condition (large maximal bladder capacity and very low grade glomerulations after cystoscopic hydrodistention might have higher sensory receptor expressions although the inflammatory protein expressions are lower than the other subtypes of IC/BPS. Interestingly, we did not find difference among the different IC/BPS phenotypes, suggesting all IC/BPS bladders have elevated sensory receptor expressions, irrelevant the severity of inflammation. The results of this study are clinically relevant in the decision of treatment strategy for patients with IC/BPS. (Lines 80-88)
The Materials and Methods section is inadequately written, and the main method of the manuscript is missing.
Reply: Thank you for the comment. We have added the methods of immunohistochemistry study and western blotting in the Methods section. (Lines 326-382)
The quality of IHC is shown in Figure 1. is quite bad. It has a large amount of background staining with unambiguously positive cells.
Reply: Thank you for the comment. We apologized that the quality of figure 1 is not very good. We have used different conditions to improve the IHC of sensory receptors, however, because all bladder tissues were harvested very long time ago, therefore, some tissue might be in poor storage condition. The purpose of this IHC figure is to demonstrate that the sensory receptors are located at the urothelium of the bladder. These pictures are the presentations for each sensory receptor expressions.
The main results are shown in Fig. 2. Representative blots were also of poor quality, and it was difficult to compare samples from the left and right blot panels. GAPDH, a housekeeping protein, is overexposed and cannot be used to normalize the expression of other proteins. It is unclear how the blots were analyzed and how they obtained the data in Table 2. The full blots are not provided in the Supplementary Material. Samples from the control and IC/BPS patients should be run on the same gels for comparison. It should be at least two control samples for each gel with IC/BPS patient samples.
Reply: Thank you for the comment. We have listed the western blots of the 52 patients with IC/BPS in the figure 2. (Lines125-128) The analysis was in accordance with standard procedure provided by the manufacture’s guidance and our previous reports. (Lines 355-382)
The statistical analysis in table 3-7 is correct, although the data used in the tables are questionable, as explained before. The discussion is long, with many irrelevant details, and the main points of the manuscript have not been discussed. Unfortunately, I cannot see how this work meets the standards of a high-impact journal, such as the International Journal of Molecular Sciences.
Reply: Thank you for the comments. We apologized that the quality of this study did not meet the high standard of reviewer like you. We have tried our best to improve the quality of writing and shorten the irrelevant discussion. As we have replied earlier, this is a clinical study rather than a basic science research. The purpose of this study was to investigate the sensory receptor expressions in the urothelium of patients with IC/BPS of different clinical phenotypes. The results of this study are clinically relevant in the decision of treatment strategy for patients with IC/BPS.

Reviewer 2 Report
Line 14: the IC and /BPS abbreviations should be explained (“interstitial cystitis”?, “bladder pain syndrome”?)
Line 44: the same for the ESSIC abbreviation.
Line 63: sigma-1 is introduced abruptly, with no explanation of its nature and relevance in the context of IC/BPS.
Lines 309-310: “Six women with normal lower urinary tract function served as controls …”. The authors have not described the demographics of their subjects, but this is in our view an important aspect that should be clarified in the paper (distribution by sex, age etc).
Lines 318-319: the source of the primary antibodies should be stated.
“The mean age of patients with IC/BPS was significantly younger than the age of the controls.” Why did the authors not try to match controls to the mean age of the subjects? How could one be certain that differences seen reflect conditions and not ages? (and the number of control subjects is considerably lower – this should at least be justified).
In Figure 2, the authors claim they provide “representative western blots”: they include only 8 out of the 52 subjects and all 6 controls. How was the “representativity” been ensured? Have the authors performed western blots for all 52 subjects but only show those of 8? Or have they somehow selected 8 samples among the 52 on which they performed the western blots (WB)? In our view if they performed WB on 52 samples, they should show all of them as a supplementary material.
Lines 117-121: the r values are difficult to interpret in the absence of scatter plots (e.g. provided as supplementary material) allowing one to assess the linearity of the relationship. Also, the statistical section of the paper does not mention at all correlation: is this Pearson, Spearman or Kendall correlation? Correlation coefficients are relatively small and confidence intervals should also be provided.
Table 2 reports protein expression levels, but I do not seem to find the method used (in the Materials and methods section) to quantify protein expression (I have only seen described immunohistochemistry and WB).
Also with respect to table 2, the statistical section states only ANOVA, but this table only indicates two groups, so it is not clear what method did they used to compute p values in this case (and whether they used any correction for statistical multiplicity or not). Also, measurement units are lacking in table 2.
Table 3 uses inferential tests to assess significance of correlation indices, but the statistical methods do not clarify how statistical significance was assessed in this context.
Table 4: apparently ANOVA was used in this case, but the post-hoc test applied is not specified here, neither in the Statistical analysis subsection. Measurement units are also lacking from this table. The same holds true for tables 5-7.
No effect size was assessed and discussed by the authors. Discussions are only focused/limited to statistically significant or insignificant associations/differences, but the effect size is never discussed. It would be important to clarify the effect size, because at least for correlations, the r values are relatively low, even though significant (explaining generally 10% of the total variability or less).
Author Response
Reviewer #2:
Line 14: the IC and /BPS abbreviations should be explained (“interstitial cystitis”?, “bladder pain syndrome”?)
Reply: Thank you for the comment. We have expanded the abbreviation of IC. (Line 14)
Line 44: the same for the ESSIC abbreviation.
Reply: Thank you for the comment. We have expanded the abbreviation of ESSIC. (Line 45)
Line 63: sigma-1 is introduced abruptly, with no explanation of its nature and relevance in the context of IC/BPS.
Reply: Thank you for the comment. We have added a description of the role of sigma-1 receptor in cystitis. (Lines 64-66)
Lines 309-310: “Six women with normal lower urinary tract function served as controls …”. The authors have not described the demographics of their subjects, but this is in our view an important aspect that should be clarified in the paper (distribution by sex, age etc).
Reply: Thank you for the comment. We have added the demographics of the six control women in the Methods. Six women (aged 55 to 78 years) with genuine stress urinary incontinence and VUDS verified normal lower urinary tract function served as controls and donated their bladder mucosa specimens during the suburethral sling surgery. (Lines 320-324)
Lines 318-319: the source of the primary antibodies should be stated.
Reply: Thank you for the comment. We have added the source of the primary antibodies in the Methods. (Lines 334-344 and Lines 361-372)
“The mean age of patients with IC/BPS was significantly younger than the age of the controls.” Why did the authors not try to match controls to the mean age of the subjects? How could one be certain that differences seen reflect conditions and not ages? (and the number of control subjects is considerably lower – this should at least be justified).
Reply: Thank you for the comment. It is not easy to obtain normal bladder tissues from women with completely normal urinary tract function which was confirmed by videourodynamic study. The tissue we obtained are donated from women who received anti-incontinence surgery and bladder tissues were harvested during the surgery. We agree that we cannot be certain that the differences of sensory receptors of IC/BPS are not caused by age. However, previous study had revealed that the ageing process increases the sensory receptors and neuropeptide expressions from bladder urothelium, which results in increased afferent mechanosensitivity and increased smooth muscle contractility. We have added this statement in the limitation of the study. (Lines 281-289)
In Figure 2, the authors claim they provide “representative western blots”: they include only 8 out of the 52 subjects and all 6 controls. How was the “representativity” been ensured? Have the authors performed western blots for all 52 subjects but only show those of 8? Or have they somehow selected 8 samples among the 52 on which they performed the western blots (WB)? In our view if they performed WB on 52 samples, they should show all of them as a supplementary material.
Reply: Thank you for the comment. We have performed western blots for all 52 patients with IC/BPS. The complete data have been added in the figure 2. (Line 125-128)
Lines 117-121: the r values are difficult to interpret in the absence of scatter plots (e.g. provided as supplementary material) allowing one to assess the linearity of the relationship. Also, the statistical section of the paper does not mention at all correlation: is this Pearson, Spearman or Kendall correlation? Correlation coefficients are relatively small and confidence intervals should also be provided.
Reply: Thank you for the comment. We have added the statement of correlation methods. Pearson’s correlation was performed to analyze the correlations between sensory receptor density and the clinical symptoms, bladder conditions, inflammatory and pro-apoptotic protein expressions in the urothelium of patients with IC/BPS. (Lines 389-391) We have added a supplementary table 2S for the correlations between sensory receptors and bladder conditions and urodynamic parameters. The scattered plots of significant correlation between measured variables and the correlation coefficients are also provided in the supplementary materials. (Lines 140-142, and Lines 500-505)
Table 2 reports protein expression levels, but I do not seem to find the method used (in the Materials and methods section) to quantify protein expression (I have only seen described immunohistochemistry and WB).
Reply: Thank you for the comment. We have added the methodology of the immunohistochemistry and western blots in the Methods section. (Lines 326-381)
Also with respect to table 2, the statistical section states only ANOVA, but this table only indicates two groups, so it is not clear what method did they used to compute p values in this case (and whether they used any correction for statistical multiplicity or not). Also, measurement units are lacking in table 2.
Reply: Thank you for the comment. We have revised the statement of the statistical analysis in the Methods section. (Lines 384-394)
Table 3 uses inferential tests to assess significance of correlation indices, but the statistical methods do not clarify how statistical significance was assessed in this context.
Reply: Thank you for the comment. We have added the statistical analysis of correlation between sensory receptors and other bladder conditions and inflammatory protein expressions in the Methods section. (Lines 389-391)
Table 4: apparently ANOVA was used in this case, but the post-hoc test applied is not specified here, neither in the Statistical analysis subsection. Measurement units are also lacking from this table. The same holds true for tables 5-7.
Reply: Thank you for the comment. We have added the post hoc test in the Method section. (Lines 387-389)
No effect size was assessed and discussed by the authors. Discussions are only focused/limited to statistically significant or insignificant associations/differences, but the effect size is never discussed. It would be important to clarify the effect size, because at least for correlations, the r values are relatively low, even though significant (explaining generally 10% of the total variability or less).
Reply: Thank you for the comment. This is a retrospective analysis to investigate the trend of sensory receptors of the bladder urothelium in patients with IC/BPS of varying bladder conditions. This is a clinical study rather than a basic science research. The purpose of this study was to investigate the sensory receptor expressions in the urothelium of patients with IC/BPS of different clinical phenotypes. We collected all our previously biopsied bladder tissue for analysis. Therefore, there was no effect size was assessed in this study. We have listed this point in the limitation of the study. Nevertheless, the results of this study are clinically relevant in the decision of treatment strategy for patients with IC/BPS. (Lines 289-294)

Reviewer 3 Report
It is well-designed and interesting paper on IC/BPS. The authors focused on the expression of sensory receptors, inflammatory markers, and pro-apoptotic markers in the bladder urothelium of patients with non-Hunner’s IC (NHIC)/BPS with different cystoscopic phenotypes and histopathological subtypes.
One revealed higher levels of urothelial sensory receptors and inflammatory and pro-apoptotic proteins in the studied cohort, while the expression was not significantly different among IC/BPS bladders with different conditions.
Please note some minor comments.
Lines 286-295 please specify how and in whom the informed consent was obtained. Please specify if these were only women (n=52).
Line 299: Please comment if the biopsies analyzed were collected from treatment-naïve patients or after e.g. btx (‘including the mucosa and submucosa, during intravesical treatment, such as botulinum toxin injections or platelet-rich plasma injections, after cystoscopic hydrodistention’). This could potentially influence your results.
Line 309: ‘Six women with normal lower urinary tract function served as controls and donated their bladder mucosa specimens during surgery for non-urinary tract disease.’ Additional information concerning informed consent and Local ethics committee is needed here, as well.
Please comment the finding from table 1 as relatively high PVR in healthy individuals (control) was observed.
Author Response
Reviewer #3
It is well-designed and interesting paper on IC/BPS. The authors focused on the expression of sensory receptors, inflammatory markers, and pro-apoptotic markers in the bladder urothelium of patients with non-Hunner’s IC (NHIC)/BPS with different cystoscopic phenotypes and histopathological subtypes. One revealed higher levels of urothelial sensory receptors and inflammatory and pro-apoptotic proteins in the studied cohort, while the expression was not significantly different among IC/BPS bladders with different conditions.
Please note some minor comments.
Lines 286-295 please specify how and in whom the informed consent was obtained. Please specify if these were only women (n=52).
Reply: Thank you for the comment. Only female IC/BPS patients were included in this retrospective analysis. (Line 296)
Line 299: Please comment if the biopsies analyzed were collected from treatment-naïve patients or after e.g. btx (‘including the mucosa and submucosa, during intravesical treatment, such as botulinum toxin injections or platelet-rich plasma injections, after cystoscopic hydrodistention’). This could potentially influence your results.
Reply: Thank you for the comment. All the bladder tissue was biopsied in patients at the first-time diagnosis of IC/BPS. These patients have not been treated with any intravesical injection. (Lines 308-309)
Line 309: ‘Six women with normal lower urinary tract function served as controls and donated their bladder mucosa specimens during surgery for non-urinary tract disease.’ Additional information concerning informed consent and Local ethics committee is needed here, as well.
Reply: Thank you for the comment. These control women had participated in our previous clinical study and informed consent had been obtained. (Lines 323-324)
Please comment the finding from table 1 as relatively high PVR in healthy individuals (control) was observed.
Reply: Thank you for the comment. The data of the controls are retrieved from the videourodynamic studies before surgery. The relatively high PVR in the controls might be due to early termination of the urination during VUDS. These patients had been confirmed no bladder outlet obstruction or low detrusor contractility, and there was small PVR in the free uroflowmetry before VUDS examination

Round 2
Reviewer 1 Report
The authors have responded to all the comments. Some of the issues still remain, but considering the samples and patients they have, this is justified. I agree with this publication in its present form.
Author Response
Reviewer #1
The authors have responded to all the comments. Some of the issues still remain, but considering the samples and patients they have, this is justified. I agree with this publication in its present form.
Reply: Thank you.

Reviewer 2 Report
The manuscript has improved, I still have a couple of unsolved issues:
"a) Also with respect to table 2, the statistical section states only ANOVA, but this table only indicates two groups, so it is not clear what method did they used to compute p values in this case (and whether they used any correction for statistical multiplicity or not). Also, measurement units are lacking in table 2.
Reply: Thank you for the comment. We have revised the statement of the statistical analysis in the Methods section. (Lines 384-394)"
The Methods section even re-written still contains only the approach for multiple group comparisons (whereas in table 2 they are 2-group comparisons). Measurement units are still lacking in table 2 (nanograms? micrograms? micrograms per gram? etc).
b) "This is a retrospective analysis to investigate the trend of sensory receptors of the bladder urothelium in patients with IC/BPS of varying bladder conditions. This is a clinical study rather than a basic science research. The purpose of this study was to investigate the sensory receptor expressions in the urothelium of patients with IC/BPS of different clinical phenotypes. We collected all our previously biopsied bladder tissue for analysis. Therefore, there was no effect size was assessed in this study. We have listed this point in the limitation of the study."
I am afraid this is not convincing, because effect size is not so much related to the clinical or non-clinical setting of a study, but rather to the statistical tools employed for a specific purpose. For instance, for ANOVA appropriate effect sizes could be Eta squared (h2), Partial Eta squared (hp2), Omega squared (w2), and Intraclass correlation (rI). As long as the authors used ANOVA, they could also measure effect size through omega squared, for instance.
Author Response
Reviewer #2
The manuscript has improved, I still have a couple of unsolved issues:
- a) Also with respect to table 2, the statistical section states only ANOVA, but this table only indicates two groups, so it is not clear what method did they used to compute p values in this case (and whether they used any correction for statistical multiplicity or not). Also, measurement units are lacking in table 2.
Reply: Thank you for the comment. We have revised the statement of the statistical analysis in the Methods section. (Lines 384-394)"
The Methods section even re-written still contains only the approach for multiple group comparisons (whereas in table 2 they are 2-group comparisons). Measurement units are still lacking in table 2 (nanograms? micrograms? micrograms per gram? etc).
Reply: Thank you for the comment. We have revised the statistical analysis in the Method section. Differences in expressions of sensory receptors between IC/BPS and controls were analyzed by Student’s t-test, (Lines 407-408)
Regarding the measurement units in the table2, and 4-7, for western blots it is normal to state that the intensity (or volume) of each protein species is determined then normalized to total protein. We have added the statement in the Method section: “The intensity of each protein was determined then normalized to GAPDH as the loading control. Data are presented as mean ± standard deviation of the relative density of sensory, inflammatory, and pro-apoptotic protein expression to GAPDH, by western blotting analysis, in the bladder tissue of IC/BPS and controls”, (Lines 400-403) and also in the footnote of tables 2-7.
- b) "This is a retrospective analysis to investigate the trend of sensory receptors of the bladder urothelium in patients with IC/BPS of varying bladder conditions. This is a clinical study rather than a basic science research. The purpose of this study was to investigate the sensory receptor expressions in the urothelium of patients with IC/BPS of different clinical phenotypes. We collected all our previously biopsied bladder tissue for analysis. Therefore, there was no effect size was assessed in this study. We have listed this point in the limitation of the study."
I am afraid this is not convincing, because effect size is not so much related to the clinical or non-clinical setting of a study, but rather to the statistical tools employed for a specific purpose. For instance, for ANOVA appropriate effect sizes could be Eta squared (h2), Partial Eta squared (hp2), Omega squared (w2), and Intraclass correlation (rI). As long as the authors used ANOVA, they could also measure effect size through omega squared, for instance.
Reply: Thank you for the comment. Effect size has been calculated in the tables. The statistical method was also added in the Method section. Effect size was calculated and the values were presented in each statistical analysis of the tables. (Lines 409-414). We also comment on the small effect size for IC/BPS subgroup analysis in the limitation of the study. (Lines 306-309)
